# NLRP3 Inflammasome: From Pathophysiology to Therapeutic Target in Major Depressive Disorder

**DOI:** 10.3390/ijms24010133

**Published:** 2022-12-21

**Authors:** Bruna R. Kouba, Joana Gil-Mohapel, Ana Lúcia S. Rodrigues

**Affiliations:** 1Center of Biological Sciences, Department of Biochemistry, Universidade Federal de Santa Catarina, Florianópolis 88040-900, SC, Brazil; 2Island Medical Program, Faculty of Medicine, University of British Columbia, Victoria, BC V8P 5C2, Canada; 3Division of Medical Sciences, University of Victoria, Victoria, BC V8P 5C2, Canada

**Keywords:** antidepressants, bioactive compounds, depression, neuroinflammation, NLRP3 complex, physical exercise

## Abstract

Major depressive disorder (MDD) is a highly prevalent psychiatric disorder, whose pathophysiology has been linked to the neuroinflammatory process. The increased activity of the Nod-like receptor pyrin containing protein 3 (NLRP3) inflammasome, an intracellular multiprotein complex, is intrinsically implicated in neuroinflammation by promoting the maturation and release of proinflammatory cytokines such as interleukin (IL)-1β and IL-18. Interestingly, individuals suffering from MDD have higher expression of NLRP3 inflammasome components and proinflammatory cytokines when compared to healthy individuals. In part, intense activation of the inflammasome may be related to autophagic impairment. Noteworthy, some conventional antidepressants induce autophagy, resulting in less activation of the NLRP3 inflammasome. In addition, the fast-acting antidepressant ketamine, some bioactive compounds and physical exercise have also been shown to have anti-inflammatory properties via inflammasome inhibition. Therefore, it is suggested that modulation of inflammasome-driven pathways may have an antidepressant effect. Here, we review the role of the NLRP3 inflammasome in the pathogenesis of MDD, highlighting that pathways related to its priming and activation are potential therapeutic targets for the treatment of MDD.

## 1. Introduction

Major depressive disorder (MDD) is a highly debilitating illness of multifactorial etiology that considerably compromises quality of life [1]. There are still several gaps in our current understanding of the pathogenesis and treatment of this disorder, and currently available antidepressants have several limitations [2]. Therefore, a better understanding of the pathophysiology of MDD is of paramount importance in the search for new therapeutic targets.

Over the last decades, it has been proposed that inflammatory processes are involved in the initiation, maintenance and relapse of MDD [3]. Notably, individuals with MDD possess increased levels of proinflammatory mediators, particularly C-reactive protein (CRP), the cytokines interleukin (IL)-6 and IL-1β, and tumor necrosis factor alpha (TNF-α) [4]. In addition, chronic activation of the immune system has been associated with alterations in the function and volume of various brain regions in MDD patients, particularly the hippocampus and medial prefrontal cortex, structures which have been shown to play a role in the regulation of affective behaviors [3]. Furthermore, immune deregulation may promote the dysregulation of several neurotransmitter pathways, including the dopaminergic, serotoninergic, noradrenergic, and glutamatergic systems [3]. Particularly, proinflammatory cytokines can mitigate serotonin synthesis (which has been directly linked to the etiology of MDD), by inducing the enzyme indoleamine 2, 3 dioxygenase (IDO), which catabolizes tryptophan into kynurenine [3].

Noteworthy, the activation of the NLRP3 inflammasome in microglia plays a crucial role during neuroinflammation by synthesizing and releasing IL-1β and IL-18, thus contributing to an increase in inflammatory cytokines [5]. Therefore, neuroinflammatory pathways, and in particular those related to NLRP3, are potential therapeutic targets for the treatment of MDD [2]. In this context, this narrative review presents an overview of preclinical and clinical studies that support a role of the NLRP3 inflammasome in the pathogenesis of MDD, highlighting that pathways related to NLRP3 inflammasome represent promising strategies for the treatment of this mood disorder.

## 2. Neuroinflammation and the NLRP3 Inflammasome

Neuroinflammation is characterized by chronic immune activation originating from a disruption of the communication between brain, microbiota, immune system, and host [6]. The neuroinflammatory process can be triggered by several factors such as stress, gut dysbiosis, infections, neurodegenerative diseases, and stroke [6,7,8]. These factors are capable of initiating an inflammatory response through pathogen-associated molecular patterns (PAMPs) and/or damage-associated molecular patterns (DAMPs), which interact with pattern recognition receptors (PRRs), including Toll-like receptors (TLRs), as well as nucleotide-binding oligomerization domain-like receptors (NLRs) present in microglia [9,10].

Within the neuroinflammatory context, microglia constitute the main mediators of this process, acting as first response cells to endogenous and exogenous insults [5]. Microglia and macrophages are able to differentiate into different phenotypes. Under normal conditions, microglia assume a branched morphology and highly mobile processes for constant monitoring of the brain parenchyma (M2 phenotype). Following an insult, microglia retracts its processes, and adopts an amoeboid form (M1 phenotype) [5,11]. The M1 phenotype positively regulates immune responsive surface proteins, such as major histocompatibility complex type II (MHCII) and chemokine receptors. In addition, it promotes the transcription of genes encoding inflammatory mediators and cytokines [5,12]. A persistent insult promotes microglial reactivation, which culminates in the release of pro-inflammatory cytokines, including TNF-α, IL-1β, IL-6, and IL-18. In addition to cytokine release, this process contributes to the generation of reactive oxygen species (ROS) and reactive nitrogen species (RNS), which in turn induce neurotoxicity [12]. Noteworthy, neuronal damage culminates in the release of ATP and/or ADP, which maintain microglial activation via the P2X purinoreceptor 7 (P2X7) [13,14].

Particularly, inflammasomes play a key role in microglial activation by sensoring various pathogens and cellular derivatives associated with damage and stress [15]. Although several inflammasomes have been described, the best characterized inflammasome related to MDD is the NLR family pyrin domain containing 3 (NLRP3) inflammasome [2,16]. This inflammasome consists of a sensor NLRP3 protein, an adaptor apoptosis-associated speck-like protein containing a caspase-recruitment domain (ASC) and the effector enzyme caspase-1. The NLRP3 protein is composed of a nucleotide-binding NACHT domain containing ATPase activity, an amino-terminal pyrin (PYD) domain, and a carboxy-terminal leucine-rich repeat (LRR) domain [16].

Activation of NLRP3 in macrophages or microglia may occur by the canonical pathway via two sequential signals: priming and activation [17]. The priming step occurs through a first stimulus promoted by ligands for TLRs, NLRs or cytokine receptors, which induces a morphological alteration into the active phenotype of these cells and promotes the expression of NLRP3 and pro-IL-1β via factor nuclear kappa B (NF-kB) and myeloid differentiation primary response 88 (Myd88) pathways [5,17]. This step also involves post-translational modifications by phosphorylation and ubiquitination of the NLRP3 protein required for inflammasome activation [15,17]. In addition, the initiation step promotes the association of NLRP3, ASC and procaspase-1 to form the inflammasome complex through PYD-PYD interactions between NLRP3 and ASC as well as CARD-CARD (caspase activation and recruitment domains) interactions between ASC and procaspase-1 [18]. Since in this step the proteins form only an inactive NLRP3 oligomeric complex, a second step called activation is required [5]. This second step is triggered by several stimuli that culminate in different cellular and molecular signaling events, such as mitochondrial dysfunction, lysosomal damage, calcium and potassium ion flux, and ROS synthesis [17]. These events activate the NLRP3 inflammasome, leading to autoproteolytic cleavage of procaspase-1 into active caspase-1 [17]. Activated caspase-1, in turn, cleaves pro-IL-18 and pro-IL-1β into the biologically active cytokines IL-18 and IL-1β, respectively [5,19].

In addition to canonical NLRP3 inflammasome activation, a non-canonical pathway may also lead to IL-1β and IL-18 synthesis through NLRP3 inflammasome activation. This may occur through binding of lipopolysaccharide (LPS) to caspases 4/5 in humans or caspase-11 in mice through a process that is independent of TLR [5,17,20]. In addition, monocytes and dendritic cells promote activation of caspase-1 and induce IL-1β secretion, without requiring a secondary stimulus through an alternative pathway [17,21,22]. Once activated, these interleukins undergo exocytosis and bind to their respective receptors IL-1R and IL-18R, which are expressed by glial cells, subsequently inducing the synthesis and release of cytokines and thus leading to an increase in the levels of IL-1β, IL-6 and TNF-α, as seen in MDD [4,5,23,24]. The neuroinflammation-induced increase in IL-1β, TNF-α, and interferon-gamma (IFN-γ) levels positively regulates the IDO enzyme, which in turn increases the conversion of tryptophan into neurotoxic metabolites such as 3-hydroxyquinurenine, 3-hydroxyanthralinic acid and quinolinic acid. Consequently, this results in a decrease in serotonin levels, which is thought to underlie, at least in part, the pathophysiology of MDD [25].

Further to the synthesis and release of cytokines, it has also been shown that the activation of NLRP3 via canonical and non-canonical pathways can lead to gasdermin D-mediated membrane pore formation (GSDMD) and subsequently pyroptosis, characterized by cell swelling and lytic cell death, which culminates in the release of intracellular DAMPs into the extracellular medium [5,18]. A schematic summary of the main events related to the activation of the NLRP3 inflammasome pathway is illustrated in Figure 1.

Microglia activation also results in intense astrocytic activation [26]. Chronic activation of these glial cells reduces the synthesis of communicating junctions that form the blood–brain barrier (BBB) and increases the levels of chemokines, such as the C-C motif chemokine ligand 2 (CCL2). This leads to the infiltration of monocytes and macrophages from the circulation into the CNS, contributing to maintain a proinflammatory state [27,28].

Of note, neuroinflammation has also been shown to be closely related with both oxidative and nitrosative stress by promoting an imbalance between the antioxidant system and the production of these reactive oxygen and nitrogen species [29,30]. This redox state impairment may result in covalent modifications to lipids, proteins, and deoxyribonucleic acid (DNA), which have been frequently reported as pathological changes in MDD [31]. Indeed, it has been noted that patients with this mood disorder display a significant increase in serum superoxide dismutase (SOD; an antioxidant enzyme) and malondialdehyde (MDA; a marker of lipid peroxidation) levels, as well as a decrease in plasma levels of ascorbic (a non-enzymative antioxidant compound). Furthermore, treatment with antidepressants such as fluoxetine and citalopram are capable of reversing these biochemical changes [32]. Considering the numerous harmful mechanisms triggered by neuroinflammation, pathways that attenuate this process are fundamental to maintaining homeostasis.

The induction of autophagic pathways has been shown to be crucial for the attenuation of the neuroinflammatory process, since they can remove and degrade DAMPs, PAMPs, NLRP3 components, and cytokines [33,34]. This process begins with the emergence of an isolated membrane, called a phagophore, which is capable of capturing cytosolic components, forming a double-membrane vesicle, called an autophagosome. Subsequently, this autophagosome fuses with lysosomes leading to the degradation and recycling of cytosolic components via lysosomal hydrolases. A network of proteins regulates this autophagic process, including Beclin-1 and LC3 (microtubule-associated protein 1 light chain 3), which are widely used as autophagic markers [34,35,36]. Interestingly, studies have shown that alterations in the autophagic process may be related to increased inflammatory cytokines in animal models of depression [37,38]. In agreement with this, a recent study showed that four autophagy-related genes (PDK4, NRG1, EphB2, and GPR18) are differentially expressed in MDD. Furthermore, these genes were significantly correlated with immune cells, suggesting an interaction between autophagy and the immune response in MDD [39]. Within this context, it has been suggested that there is intricate communication between the NLRP3 inflammasome and the autophagy system. Indeed, caspase-1 activation may promote increased inflammasome activation while also inhibiting autophagy via cleavage and a consequent decrease in the levels of the signaling intermediate Toll/IL-1R domain-containing adaptor-inducing IFN-β (TRIF) [40,41]. This evidence suggests that crosstalk between NLRP3 activation and altered autophagy may contribute to the pathophysiology of MDD [34].

## 3. Involvement of NLRP3 Inflammasome in MDD

### 3.1. Preclinical Evidence

Zhang et al. (2014) conducted the first study to assess inflammasome activation in a mouse model of depression. In this study, lipopolysaccharide (LPS)-treated male BALB/c mice (0.8 mg/kg, i.p.) exhibited an increase in NLRP3 inflammasome mRNA expression, resulting in an increase of proinflammatory cytokine IL-1β mRNA and protein levels in the brain. These changes were accompanied by the induction of depressive-type and anhedonic-type behaviors. In addition, pretreatment with the NLRP3 inflammasome inhibitor Ac-Tyr-Val-Ala-Asp-chloromethylketone (8 mg/kg, i.p.) was capable of mitigating these LPS-induced depressive-like behaviors [42].

In a different study, depressive-like behaviors in male C57BL/6 mice were observed up to 29 days after a single LPS administration (5 mg/kg, i.p.) in a manner dependent on the activation of the NLRP3 inflammasome. It was found that three days after LPS administration, the expression of NLRP3, ASC, caspase-1 p10, TNF-α IL-1β, and IL-18 in the hippocampus and the levels of IL-1β and TNF-α in the serum remained elevated, while hippocampal IL-10 levels decreased as a result of microglial activation [43]. On the other hand, repeated daily administration of increasing doses of LPS (0.1, 0.42 and 0.83 mg/kg; i.p.) has been shown to induce depressive-like behaviors in C57BL/6J mice by a mechanism independent of NLRP3 inflammasome activation [44].

Data from various stress-exposed animal models of depression also suggest that there is a relationship between the development of depressive-like behaviors and the NLRP3 inflammasome [45,46]. Male BALB/c mice subjected to a chronic unpredictable mild stress (CUMS) protocol for 4 weeks displayed depressive-like behaviors and increased hippocampal levels of active IL-1β [46]. In a different study, rats subjected to 6 h of restraint stress for 21 days exhibited hippocampal alterations characterized by increased Iba-1 expression, ROS formation, NF-kB, NLRP3, cleaved caspase-1, IL-1β and IL-18 levels [45].

A study by Pan et al. (2014) observed that male Wistar rats exposed to CUMS for a period of 12-weeks over-expressed P2X7 and TLR2 in the prefrontal cortex, causing activation of the NF-kB pathway and consequently the NLRP3 inflammasome, resulting in microglial activation, astrocytic impairment, and increased prefrontal cortex IL-1β expression [47]. The involvement of the P2X7 receptor in inflammasome activation was further observed in male Sprague Dawley rats submitted to CUMS for 3 weeks. In this study, CUMS induced depressive-like behaviors through a mechanism dependent on the P2X7 receptor. In addition, rats subjected to chronic unpredictable stress (CUS) exhibited increased levels of extracellular ATP, cleaved-caspase 1, ASC, and IL-1β in the hippocampus [13].

Following NLRP3 inflammasome activation, other events including pyroptosis, IDO activation and alterations of the autophagic process have also been associated with the induction of depressive-like behaviors [48,49,50]. Indeed, it has been recently shown that administration of monosodium glutamate results in an increase in the levels of Gasdermin D (GSDMD; a protein that induces pyroptosis), caspase-1, IL-1β, IL-18, NLRP3, and ASC in the hippocampus of postnatal rats [50].

LPS administration (1.8 mg/kg, i.p.) has been shown to increase the expression and activity of IDO in the hippocampus while also inducing depressive-like behaviors in C57BL/6 mice in an NLRP3-dependent manner [51]. Similarly, male Sprague Dawley rats submitted to CUMS for 4 weeks exhibited increased activation of IDO, which caused an increase in the kynurenine/tryptophan ratio and a consequent reduction in serotonin levels. In addition, these animals developed depressive-like behaviors through the activation of the P2X7/NLRP3 inflammasome axis, resulting in increased IL-1β, IL-6, and TNF-α expression [52].

The administration of LPS (500 μg/kg, i.p., every 2 days for a total of seven injections) has been shown to suppress autophagic markers, including LC3II/LC3I and beclin-1, while also inducing neuroinflammation via NLRP3, and this seems to be associated with the occurrence of depressive-like behaviors in rats [37]. Wang et al. (2020) also showed that LPS increased the expression of pro-inflammatory cytokines and caused NLRP3 inflammasome activation paralleled with inhibition of autophagy in the hippocampus of rats and in BV2 cells [53]. In male rats subjected to restraint stress (6-hour for 28 days), the development of depressive-like behaviors was accompanied by a concomitant dysfunction of the AMPK-mTOR (AMP-activated protein kinase-mammalian target of rapamycin) pathway and a decreased expression of LC3-II and beclin-1 in the prefrontal cortex [49]. On the other hand, administration of corticosterone to male mice (20 mg/kg, p.o., for 21 days) was shown to induce the development of depressive-like behaviors without altering autophagy-related proteins phospho-mTORC1, LC3A/B, and beclin-1 in the hippocampus [54].

The microbiota-gut-inflammasome-brain axis is a bidirectional communication system linking psychological stress responses, immune system function, and gut microbiome composition [55,56]. Within this context, it has been suggested that stress-induced increases in NLRP3 signaling can in turn promote pro-inflammatory bacterial clades within the gut microbiota composition. These microbiota changes may alter gut barrier function and result in the increased translocation of bacteria to otherwise sterile enteric compartments, thus contributing to the inflammatory process. Furthermore, dysbiosis can compromise the bioavailability of monoamines and neuroactive compounds, further exacerbating depressive symptoms [55]. In line with this idea, a recent study has shown that transplantation of fecal microbiota from control male Sprague Dawley rats ameliorated the occurrence of depressive-like behaviors in CUMS-exposed rats. Furthermore, fecal microbiota transplantation restored the levels of serotonin and suppressed the activation of microglial and astrocytic cells, reducing the expression of NLRP3, ASC, caspase-1, and IL-1β in the hippocampus of CUS-exposed animals [57,58]. Reinforcing the existence of a link among inflammation, microbiota, and depression, transplantation of fecal microbiota from NLRP3 KO mice has been shown to alleviate chronic stress-induced depressive-like behaviors in recipient mice. In addition, fecal microbiota transplantation also attenuated astrocytic activation [59]. These findings support the notion that stress- or LPS-triggered neuroinflammation can induce depressive-like behaviors by promoting the activation of the NLRP3 inflammasome and that modulation of gut microbiota may be a critical target in alleviating NLRP3-driven neuroinflammation. Figure 2 illustrates the main pathophysiological events underlying the neuroinflammatory process associated with the activation of the NLRP3 inflammasome and the development of depressive-like behaviors in animal models.

### 3.2. Clinical Evidence

Alcocer-Gómez et al. (2014) has reported a correlation between levels of IL-1β and IL-18 and Beck Depression Inventory (BDI) scores. Specifically, MDD patients showed enhanced expression of NLRP3 and caspase-1 in blood cells, resulting in increased serum levels of IL-1β and IL-18 [60]. Similarly, a recent study showed that NLRP3 and caspase-1 mRNA levels were significantly elevated in MDD patients when compared with healthy controls [61].

A study conducted with postmortem brains of individuals afflicted with MDD that died by suicide assessed the involvement of other NLRPs inflammasomes in depression. In this study, protein and mRNA expression levels of ASC, NLRP3, NLRP1, NLRP6, and caspase-3 were shown to be significantly increased in the prefrontal cortex of MDD individuals when compared to healthy controls. In addition, an increase in both mRNA and protein levels of IL-1β, TNF-α and IL-6 was also observed [62]. As such, future studies are warranted to further elucidate the involvement of other inflammasomes in MDD.

In other studies, NLRP3 inflammasome-induced increases in the levels of several inflammatory such as IL-1β, IL-6, and TNF-α have also been observed in individuals with MDD [63,64]. However, there are some inconsistencies between studies regarding the observed increases of cytokines [65,66]. For example, while one meta-analysis study reported significantly higher concentrations of TNF-α and IL-6 in MDD afflicted individuals when compared to healthy controls [65], another meta-analysis study observed no consistent association between TNF-α and IL-1β levels and MDD [66]. Therefore, more studies are needed to clearly evaluate the correlation between inflammatory cytokines and MDD in humans.

## 4. Anti-Inflammatory Effects of Antidepressants

A compilation of studies has reported that antidepressant drugs may exhibit anti-inflammatory and autophagic effects. Tricyclic antidepressants, including clomipramine, citalopram and imipramine have anti-inflammatory effects by inhibiting IL-6, IL-1β and TNF-α release in human monocytes [67]. The administration of clomipramine (20 mg/kg for 1 week, i.p.) is able to decrease the expression of IL-1β, IL-6, and TNF-α without changing the levels of NLRP3 in C57BL/6 male injected with LPS. However, in BV2 cells, this antidepressant was shown to reduce the increase in NLRP3 gene expression [68]. In another study, clomipramine (20 mg/kg for 1 week, i.p.) did not promote alterations in the expression of caspase-1 and NLRP3, but was still able to ameliorate depressive-like behaviors in LPS-treated male C57BL/6J mice by regulating the expression of ASC and IDO [69].

Fluoxetine was also reported to reduce IFN-γ and TNF-α levels and decrease the IFN-γ/IL-10 ratio in whole blood from control volunteers [70]. Moreover, the anti-inflammatory effect of chronic fluoxetine treatment was shown to be related to NLRP3 inflammasome modulation in the prefrontal cortex in a rat model of depression [47]. In this study fluoxetine (10 mg/kg for 6 weeks, i.p.) inhibited NF-kB pathway activation, decreased NLRP3 protein, and increased IL-1β levels in the prefrontal cortex of male Wistar rats exposed to chronic unpredictable mild stress (CUMS) [47]. Fluoxetine (10 mg/kg for 4 weeks) was also shown to suppress NLRP3 inflammasome activation, subsequent caspase-1 cleavage, and IL-1β secretion in hippocampal microglia from male C57BL/6 mice subjected to stress [71]. Similarly, in the hippocampus of male C57BL/6J mice, fluoxetine treatment (20 mg/kg, i.g.) reversed LPS-induced changes be decreasing the expression of NLRP3, TNF-α, IL-1β, IL-6, and caspase-1, and increasing the levels of the IL-10 anti-inflammatory cytokine [72]. In addition, it was shown that the decrease in the expression of TLR4, NF-kB, NLRP3, caspase-1, TNF-α and IL-1β caused by fluoxetine treatment (10 mg/kg/day, p.o.) attenuated the progression of Alzheimer-like phenotypes in socially isolated depressed-like rats [73].

Noteworthy, clinical studies highlight that treatment with antidepressants, including fluoxetine, paroxetine, mianserin, mirtazapine, venlafaxine, desvenlafaxine, amitriptyline, imipramine and agomelatine is associated with induction of autophagy and a decrease in the expression of NLRP3 inflammasome components as well as IL-1β and IL-18 inflammatory cytokines [60,74].

In addition to conventional antidepressants, ketamine has also been shown to have anti-inflammatory effects [75]. A sub-anesthesia dose of ketamine (10 mg/kg, i.p.) was shown to decrease IL-1β and NLRP3 expression in the hippocampus while also ameliorating LPS-induced depressive-like behaviors in male C57BL/6 mice [75]. Similarly, the anti-inflammatory effect of ketamine has also been observed in a chronic restraint stress model of depression, inducing a reduction in the expression of NLRP3 inflammasome-related proteins in this model. In this study, a sub-anesthetic dose of this antidepressant was able to trigger autophagy in the hippocampus and prefrontal cortex of Wistar Kyoto rats [76]. Overall, these studies suggest that attenuating the neuroinflammatory process may be critical to the treatment of MDD.

## 5. Antidepressant and Anti-Inflammatory Effects of Bioactive Compounds

Several bioactive compounds with anti-inflammatory properties have been shown to also possess antidepressant effects in animal models (Table 1).

Baicalin, a polyphenol, was shown to have anti-inflammatory properties by decreasing IL-1β levels in the prefrontal cortex and hippocampus of male rats submitted to CUMS [77,78]. Curcumin was also shown to inhibit the P2X7/NLRP3 inflammasome axis, reducing the CUMS-induced conversion of pro-IL-1β to mature IL-1β in the hippocampus of male Sprague Dawley rats [52].

The flavonoid apigenin attenuated the expression of NLRP3 and IL-1β caused by CUMS in the prefrontal cortex of rats [79]. In addition, the flavonoid astragalin also reduced the expression of NLRP3 and IL-1β while also decreasing the protein levels of NF-kB p65, NLRP3, capase-1, gasdermin D, and IL-1β in the hippocampus of female C57BL/6 mice exposed to CUMS [80]. Similarly, the administration of paeoniflorin (20 and 40 mg/kg, o.g.) was able to inhibit the enhanced expression of the pore-forming protein gasdermin D in the hippocampus of male C57BL/6 mice treated with reserpine [81]. This monoterpene has also been shown to inhibit TLR4/NF-kB/NLRP3 signaling, decreasing IL-β levels and microglial activation in the hippocampus of male Institute of Cancer Research (ICR) mice treated with LPS [82]. In addition, the licorice-derived flavonoid isoliquiritin was shown to suppress NLRP3-mediated pyroptosis via the miRNA-27a/SYK/NF-kB axis in both the LPS and the chronic social defeat stress models of depression [83].

Salvianolic acid B has also been shown to mitigate the expression of NLPR3, ASC and caspase-1 in LPS- and CUMS-exposed rats [37,84]. Moreover, this compound can promote autophagy, thereby inducing NLRP3 clearance, and this appears to be related with its antidepressant properties [37]. Similarly, the polymethoxylated flavonoid nobiletin was also shown to attenuate NLRP3 inflammasome activation by promoting autophagy in LPS-treated male Sprague Dawley rats [53].

Finally, sulforaphane, an isothiocyanate found in cruciferous vegetables that is and activator of the Nrf2 antioxidant pathway, also exhibits antidepressant-like and anti-inflammatory properties [85]. Indeed, sulforaphane was shown to inhibit NLRP3 inflammasome activation and suppress oxidative stress and pyroptotic cell death caused by LPS and ATP in a murine N9 microglial cell line [86].

## 6. Inhibition of NLRP3 Inflammasome by Physical Exercise

Physical exercise is known to possess antidepressant effects both in humans and rodents [87,88,89,90,91]. The mechanisms underlying its antidepressant effects have been extensively studied and are thought to involve neurogenic stimulation by increasing the expression of brain-derived neurotrophic factor (BDNF) in critical brain regions, particularly in the dentate gyrus of the hippocampus [89,92]. More recently, modulation of the NLRP3 inflammasome by physical exercise has also received attention. A study by Abkenar et al. (2019) reported that plasma expression of NLRP3 as well as IL-1β and 1L-18 is decreased following moderate-intensity aerobic training in young men, whereas chronic high-intensity running resulted in activation of the NLRP3 inflammasome [93].

Numerous studies have also demonstrated the antidepressant-like effects of physical exercise in rodents [90,91,92,94], while recent reports have also shown that physical activity can inhibit NLRP3 activation in the hippocampus [89,95]. For example, in a study by Wang et al. (2016) exercise was shown to attenuate the occurrence of depressive-like behaviors in ovariectomized female rats, while also decreasing hippocampal levels of IL-1β and IL-18 via suppression of NLRP3 inflammasome activation [94]. In a different study using a post-stroke depression model, physical exercise was also shown to block the TLR4/NF-kB/NLRP3 signaling pathway in the mouse hippocampus through inhibition of the phosphatase and tensin homologue (PTEN) [96]. Moreover, some evidence also suggested that physical exercise may be a possible non-pharmacological strategy to prevent inflammatory events in mice treated with Aβ_1–40_ by preventing hippocampal activation of the NLRP3 inflammasome [89] as well as gut dysfunction [90].

## 7. Conclusions and Future Directions

In this review, we discussed evidence supporting the existence of an intrinsic relationship between the NLRP3 inflammasome and the development of MDD. Indeed, preclinical and clinical studies have repeatedly demonstrated that an increase in the expression and activation of this inflammasome induces a higher release of cytokines, such as TNF-α, IL-1β, and IL-6, which have been shown to be elevated in individuals with this mood disorder. Although conventional antidepressants have several limitations, including modest efficacy rates, clinical and preclinical studies demonstrate that these drugs have anti-inflammatory effects that are mediated, at least partially, through inhibition of the NLRP3 inflammasome. Interestingly and as discussed above, ketamine is also able to inhibit the expression and activation of this complex. Indeed, a recent preclinical study demonstrated that a single administration of ketamine prior to LPS treatment or TNF-α administration was able to prevent the increase of NLRP3 inflammasome complex components, producing a pro-resilience effect against the development of depressive-like behaviors [97]. Therefore, it is plausible that the specific modulation of pathways related to the inflammasome complex may have improved therapeutic efficacy. Future studies are thus warranted to further explore the mechanisms related to the anti-inflammatory effects of these drugs, especially with regard to autophagic induction. In addition, natural bioactive compounds and physical exercise may also be useful in the management of MDD and their antidepressant properties also appear to be mediated, at least in part, by the inhibition of the NLRP3 inflammasome. Considering that MDD is projected to be the leading cause of disease burden worldwide by 2030 [1], the anti-inflammatory strategies reviewed here deserve further research, so as to completely elucidate their full therapeutic potential in the context of MDD.

## Figures and Tables

**Figure 1 ijms-24-00133-f001:**
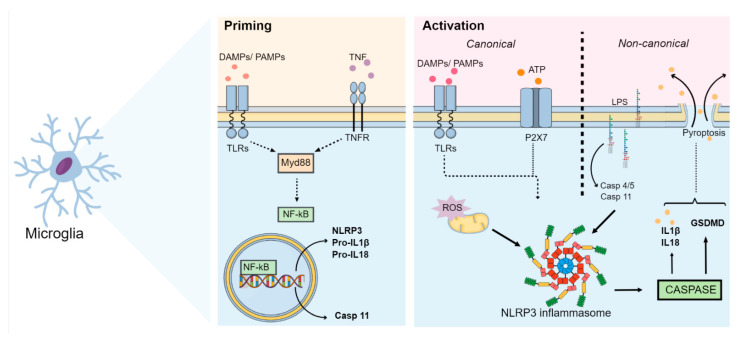
NLRP3 inflammasome pathways. After an insult, the NLRP3 inflammasome initiation step occurs. During this step, DAMPs, PAMPs and cytokines (such as TNF-α) interact with TLRs, NLRs or cytokine receptors (such as TNFR). This interaction induces the activation of the Myd88 and NF-kB pathways. Activated NF-kB then translocates to the nucleus and induces the expression of NLRP3, pro-IL-1β, pro-IL-18 and caspase-11 (through the non-canonical pathway). During this activation step, several stimuli culminate in different cellular and molecular signaling events, such as mitochondrial dysfunction, lysosomal damage, calcium and potassium ion flux, ROS synthesis, and ATP release. These events activate the NLRP3 inflammasome, which cleaves procaspase-1 and releases the active form of caspase-1. Caspase-1 then cleaves pro-IL-18 and pro-IL-1β into their active forms and activates GSDMD, thus inducing pyroptosis. Synthesis of IL-1β and IL-18 as well as activation of GSDMD by NLRP3 may also occur via a non-canonical pathway, in which LPS binds directly to caspases 4/5 in humans and caspase-11 in mice. Abbreviations: DAMPs: damage-associated molecular patterns; GSDMD: gasdermin D-mediated membrane pore formation; IL: interleukin; LPS: lipopolysaccharide; Myd88: myeloid differentiation primary response 88; NF-kB: nuclear factor kappa B; NLR: nucleotide-binding oligomerization domain-like receptor; NLRP3: Nod-like receptor pyrin containing 3; PAMPs: pathogen-associated molecular patterns; TLRs: Toll-like receptors; TNF-α: tumor necrosis factor; TNFR: tumor necrosis factor receptor.

**Figure 2 ijms-24-00133-f002:**
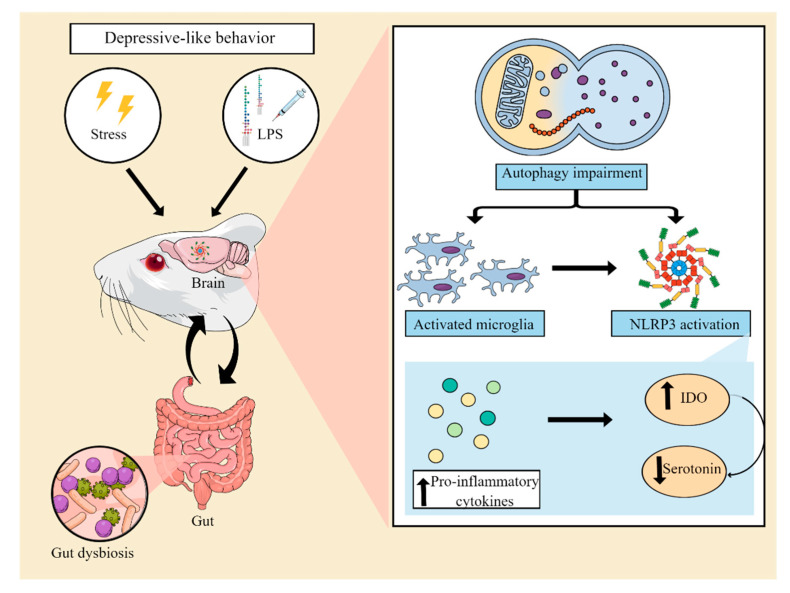
Involvement of the NLRP3 inflammasome in preclinical models of depression. Animal models of depression induced by stress or an inflammatory challenge with lipopolysaccharide (LPS) have been used to elucidate the mechanisms involved in the pathophysiology of MDD. These models are able to induce a neuroinflammatory process, associated with an impairment of autophagic pathways and microglial activation. This process leads to the assembly and activation of the NLRP3 complex. As a consequence, there is an increase in the levels of proinflammatory cytokines, which in turn result in increased indoleamine 2,3 dioxygenase (IDO) activity, decreased tryptophan bioavailability, and a consequent reduction in serotonin synthesis. Furthermore, NLRP3 activation has also been suggested to be associated with gut dysbiosis, compromising the production of serotonin and neuroprotective compounds by the gut microbiota. Abbreviations: NLRP3: Nod-like receptor pyrin containing protein 3.

**Table 1 ijms-24-00133-t001:** Antidepressant and anti-inflammatory effects of bioactive compounds.

Bioactive Compound	Animal Model	Behavioral Alterations	Biochemical Alterations	References
Baicalin (20 and 40mg/kg; o.g.)	Male Sprague Dawley rat submitted to CUMS	Antidepressant-like effect in the FST and SPT	↓ ASC, NLRP3, caspase-1, and IL-1β	[77,78]
Curcumin (100 mg/kg; o.g.)	Male Sprague Dawley rats submitted to CUMS	Antidepressant-like effect in the FST and SPT	↓ P2X7/NLRP3 inflammasome axis, pro-IL-1β, and IL-1β	[52]
Apigenin (20 mg/kg; i.p.)	Male Sprague Dawley rats submitted to CUMS	Antidepressant-like effect in the SPT	↓ NLRP3 and IL-1β	[79]
Astragalin (10 mg/kg)	Female C57BL/6 mice submitted to CUMS	Antidepressant-like effect in the TST and SPT	↓ NF-kB p65, NLRP3, capase-1, gasdermin D, and IL-1β	[80]
Paeoniflorin (20 and 40 mg/kg, o.g.)	Male C57BL/6 mice treated with reserpine	Antidepressant-like effect in the TST and FST	↓ Gasdermin D	[81]
Paeoniflorin (20, 40 and 80 mg/kg, o.g.)	Male ICR mice submitted to LPS	Antidepressant-like effect in the FST	↓ TLR4/NF-kB/NLRP3 signaling, IL-β, IL-6, and TNF-α levels	[82]
Isoliquiritin (10 and 30 mg/kg)	Male C57BL6/J mice submitted to LPS and chronic social defeat stress	Antidepressant-like effect in the SPT, TST, and FST	↓ Pyroptosis	[83]
Salvianolic acid B	Male rats submitted to LPS and CMS	Antidepressant-like effect in the SPT, TST, and FST	↓ NLPR3, ASC and caspase-1	[37,84]
Salvianolic acid B (20 mg/kg, i.p)	Male, Sprague Dawley rats submitted to LPS	Antidepressant-like effect in the SPT, and FST	Promote autophagy, and ↓ NLRP3	[37]
Nobiletin (100 mg/kg, i.p.)	Male Sprague Dawley rats submitted to LPS	Antidepressant-like effect in the SPT, and FST	Promote autophagy (↑ Beclin 1, LC3) ↓ ASC, caspase-1, and IL-1β	[53]

Abbreviations: ASC, apoptosis-associated speck-like protein containing a caspase recruitment domain; CMS, chronic mild stress; CUMS, chronic unpredictable mild stress; FST, forced swimming test; ICR, Institute of Cancer Research; IL, interleukin; LPS, lipopolysaccharide; NF-kB, nuclear factor kappa B; NLRP3, Nod-like receptor pyrin containing 3; SPT, sucrose preference test; TLR4, Toll-like receptor 4; TST, tail suspension test; ↓, decreased; ↑, increased.

## Data Availability

Not applicable.

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
