# Peer review of "NLRP3 Inflammasome: From Pathophysiology to Therapeutic Target in Major Depressive Disorder"

_ijms, 2022, doi:10.3390/ijms24010133_

Round 1

Reviewer 1 Report

Manuscript entitled " NLRP3 inflammasome: from pathophysiology to therapeutic target in major depressive disorder" is very usefull, because it unifay knowledge about inflammation in depressive disorder.

I have just minor suggestion. Instead of figure 2, which is not needed, it would be more clear to present some of pathophysiological process as a figure, sheme or table in manuscipt.

In addition, considering that inflammation is connected very tightly to oxidative stress, did you find in the literature how oxidative stress as a consequence of inflammation influences major depression. Also, do antidepressant drugs have influence on redox status in these patients?

Author Response

Reviewer #1: Manuscript entitled "NLRP3 inflammasome: from pathophysiology to therapeutic target in major depressive disorder" is very useful, because it unifies knowledge about inflammation in depressive disorder.

Comment: We thank the reviewer for their positive feedback.

  1. I have just minor suggestion. Instead of figure 2, which is not needed, it would be more clear to present some of pathophysiological process as a figure, scheme or table in manuscript.

Response: Please note that we have reformulated Figure 2 as suggested by the Reviewer. The revised Figure 2 and corresponding Figure Legend briefly outline the pathophysiological processes associated with NLRP3 activation in animal models of depression, with the Figure Legend reading as follows:

Involvement of the NLRP3 inflammasome in preclinical models of depression. Animal models of depression induced by stress or an inflammatory challenge with lipopolysaccharide (LPS) have been used to elucidate the mechanisms involved in the pathophysiology of MDD. These models are able to induce a neuroinflammatory process, associated with an impairment of autophagic pathways and microglial activation. This process leads to the assembly and activation of the NLRP3 complex. As a consequence, there is an increase in the levels of proinflammatory cytokines, which in turn result in increased indoleamine 2,3 dioxygenase (IDO) activity, decreased tryptophan bioavailability, and a consequent reduction in serotonin synthesis. Furthermore, NLRP3 activation has also been suggested to be associated with gut dysbiosis, compromising the production of serotonin and neuroprotective compounds by the gut microbiota”. 

  1. In addition, considering that inflammation is connected very tightly to oxidative stress, did you find in the literature how oxidative stress as a consequence of inflammation influences major depression. Also, do antidepressant drugs have influence on redox status in these patients?

Response: We thank the Reviewer for raising this interesting issue. Please note that we have now added this information to the manuscript. The new paragraph reads as follows:

Of note, neuroinflammation has also been shown to be closely related with both oxidative and nitrosative stress by promoting an imbalance between the antioxidant system and the production of these reactive oxygen and nitrogen species [29, 30]. This redox state impairment may result in covalent modifications to lipids, proteins, and deoxyribonucleic acid (DNA), which have been frequently reported as pathological changes in MDD [31]. Indeed, it has been noted that patients with this mood disorder display a significant increase in serum superoxide dismutase (SOD; an antioxidant enzyme) and malondialdehyde (MDA; a marker of lipid peroxidation) levels, as well as a decrease in plasma levels of ascorbic (a non-enzymative antioxidant compound). Furthermore, treatment with antidepressants such as fluoxetine and citalopram are capable of reversing these biochemical changes [32]. Considering the numerous harmful mechanisms triggered by neuroinflammation, pathways that attenuate this process are fundamental to maintaining homeostasis”.

Reviewer 2 Report

This manuscript discussed the role of NLRP3 inflammasome in major depressive disorder and the anti-inflammatory pharmacological and non-pharmacological strategies were suggested in conclusions. Overall, this review has a lot of qualities, it is well designed, the text is comprehensive and generally well organized into sections. New and significant findings have been highlighted in the manuscript with the directions for future research. Schematic presentations are also clear and make the text to be followed easier.

In my opinion, there are only few concerns for the manuscript modification.

Although the text is comprehensive, I think that microbiota‐inflammasome interactions should be more in-depth explained in the text. There is one review on this topic that might be useful: Inserra A, Rogers GB, Licinio J, Wong ML. The microbiota‐inflammasome hypothesis of major depression. Bioessays. 2018 Sep;40(9):1800027. Besides, there is another manuscript in this research filed that deserves to be mentioned: Yang F, Zhu W, Cai X, Zhang W, Yu Z, Li X, Zhang J, Cai M, Xiang J, Cai D. Minocycline alleviates NLRP3 inflammasome-dependent pyroptosis in monosodium glutamate-induced depressive rats. Biochemical and biophysical research communications. 2020 Jun 4;526(3):553-9.

The terms ‘CUMS’ and ‘CUS’ should be defined at the first times mentioned in the text (lines 185 and 197).

Keywords other from those used in the title should be stated after the abstract.

Author Response

Reviewer #2: This manuscript discussed the role of NLRP3 inflammasome in major depressive disorder and the anti-inflammatory pharmacological and non-pharmacological strategies were suggested in conclusions. Overall, this review has a lot of qualities, it is well designed, the text is comprehensive and generally well organized into sections. New and significant findings have been highlighted in the manuscript with the directions for future research. Schematic presentations are also clear and make the text to be followed easier.

Comment: We thank the reviewer for their positive feedback. 

  1. Although the text is comprehensive, I think that microbiota‐inflammasome interactions should be more in-depth explained in the text. There is one review on this topic that might be useful: Inserra A, Rogers GB, Licinio J, Wong ML. The microbiota‐inflammasome hypothesis of major depression. Bioessays. 2018 Sep;40(9):1800027. Besides, there is another manuscript in this research filed that deserves to be mentioned: Yang F, Zhu W, Cai X, Zhang W, Yu Z, Li X, Zhang J, Cai M, Xiang J, Cai D. Minocycline alleviates NLRP3 inflammasome-dependent pyroptosis in monosodium glutamate-induced depressive rats. Biochemical and biophysical research communications. 2020 Jun 4;526(3):553-9.

Response: We thank the Reviewer for pointing out these additional publications. As suggested, we have now cited and discussed these studies in the revised version of our manuscript. The new paragraphs read as follows:

Following NLRP3 inflammasome activation, other events including pyroptosis, IDO activation and alterations of the autophagic process have also been associated with the induction of depressive-like behaviors [48-50]. Indeed, it has been recently shown that administration of monosodium glutamate results in an increase in the levels of Gasdermin D (GSDMD; a protein that induces pyroptosis), caspase-1, IL-1β, IL-18, NLRP3, and ASC in the hippocampus of postnatal rats [50]”.

“The microbiota-gut-inflammasome-brain axis is a bidirectional communication system linking psychological stress responses, immune system function, and gut microbiome composition [55, 56]. Within this context, it has been suggested that stress-induced increases in NLRP3 signaling can in turn promote pro-inflammatory bacterial clades within the gut microbiota composition. These microbiota changes may alter gut barrier function and result in the increased translocation of bacteria to otherwise sterile enteric compartments, thus contributing to the inflammatory process. Furthermore, dysbiosis can compromise the bioavailability of monoamines and neuroactive compounds, further exacerbating depressive symptoms [55]”.

  1. The terms ‘CUMS’ and ‘CUS’ should be defined at the first times mentioned in the text (lines 185 and 197).

Response: We thank the Reviewer for pointing out this omission. We have now defined these two abbreviations (CUS, chronic unpredictable stress and CUMS, chronic unpredictable mild stress) the first time they were used in the manuscript.

  1. Keywords other from those used in the title should be stated after the abstract.

Response: We thank the Reviewer for pointing out this omission. As suggested, we have now revised the list of keywords provided after the Abstract as follows:

Antidepressants; bioactive compounds; depression; neuroinflammation; NLRP3 complex; physical exercise”.
